# A Novel Procedure to Measure Membrane Penetration of Coarse Granular Materials

**Weiwei Xu** [1,2,*], **Enyue Ji** [1,2,*], **Beixiao Shi** [1,2] **and Chenghao Chen** [1,2]

1    Geotechnical Engineering Department, Nanjing Hydraulic Research Institute, Nanjing 210024, China; shibeixiao@nhri.cn (B.S.); chchen@nhri.cn (C.C.)
2    Key Laboratory of Failure Mechanism and Safety Control Techniques of Earth-Rock Dam of the Ministry of Water Resources, Nanjing 210024, China
*    Correspondence: weiweixu@live.cn (W.X.); enyueji@nhri.cn (E.J.)

**Abstract:** The membrane penetration effect will significantly influence the measurement of specimen volume deformation in the triaxial test. This paper presents a novel procedure for measuring and correcting the membrane penetration of rubber by using a newly developed multiscale triaxial apparatus. A series of triaxial tests on coarse granular materials was conducted with different specimen diameters, and it is found that the proportion of volume change due to membrane penetration decreases linearly with increasing specimen diameter. To reduce the margin of error induced by membrane penetration in the triaxial test, it is recommended to use specimens of larger size. Such a method can facilitate the correction and estimation of the membrane penetration effect of coarse granular materials.

**Keywords:** coarse granular materials; membrane penetration; triaxial tests; volume change; specimen size

## 1. Introduction

The triaxial test is a common method to obtain the stress–strain behavior of coarse granular materials in the laboratory [1]. When preparing coarse granular materials for triaxial tests, test specimen is typically covered by rubber membrane to prevent direct contact with water in the pressure chamber, so that boundary stress and control drainage conditions can be readily applied. However, due to the unevenness of the specimen, the confining pressure will cause the flexible rubber membrane to be pushed into the peripheral voids of the specimen, which is called membrane penetration [1]. This situation can be found in both conventional triaxial tests and true triaxial tests. By ignoring the impact of rubber membrane penetration, the real volume change of the specimen may be overestimated in the triaxial consolidated-drained test. Furthermore, in the undrained test, the total volume of the saturated specimen remains unchanged. When applying the confining pressure, positive pore pressure is generated in the specimen, the membrane will be withdrawn out of the peripheral voids of the soil specimen, and the effective confining pressure decreases; then, the membrane penetration will decrease with increasing pore water pressure. This leads to a certain amount of water in the soil skeleton being drained, invalidating the assumed testing conditions of a constant volume without drainage [2]. Consequently, the undrained test will become a drained test with decreasing measured pore pressure increment Δu, resulting in an unconservative overestimation of the cyclic strength of the soil in the cyclic triaxial test. Thus, the membrane penetration effect cannot be ignored, and the measurement error caused by the penetration effect must be corrected or reduced [2].

Newland and Alley [3] first recognized membrane penetration as a source of revised volume change and divided the total volume change into two parts: the volume change of the soil skeleton and the rubber membrane penetration volume. Newland and Alley

also assumed that the specimen behaved isotropically under hydrostatic loading, and the volumetric strains were calculated to be three times equivalent of the axial strains [4]. However, the strain of sand is anisotropic under hydrostatic loading, the radial strains will be larger than the axial strains, and the assumption overestimates the penetration volume change of the membrane, as explained by EI-Sohby and Andrawes, Vaid and Negussey, Kramer and Sivaneswaran [5–7].

To address the membrane penetration issue, researchers usually study membrane penetration based on theoretical derivation. Roscoe et al. proposed a method for evaluating the membrane penetration volume change. The cylindrical rigid brass rods of different diameters were placed into the center of soil specimens, then the total volume change under different confining pressures was measured [8]. A linear relationship between the total volume change and the diameter of the brass rods can therefore be plotted. Since brass rods are relatively incompressible, the membrane penetration is equal to the total volume change when the diameter of the brass rods and the specimen diameter are the same. Raju and Sadasivan [9] stated the following disadvantages of the method proposed by Roscoe et al. (1) The relationship between the total volume change and diameter of the brass rod are not linear; and (2) when the top loading platen rests on the brass rod, the vertical stress on the soil will be lower than the applied hydrostatic stress, resulting in nonhydrostatic compression of the soil skeleton. A similar procedure was adapted by Raju and Sadasivan. To ensure that the specimen was subjected to hydrostatic stress, the conventional rigid top platen was replaced with an annular flexible and lubricated top plate. JI et al. [10] adapted the above procedure to study membrane penetration, and found a good linear relationship between the total volume change and diameter of the brass rod when the confining stress was between 35 and 600 kPa. When the confining pressure exceeded 600 kPa, this relationship was no longer linear. Therefore, Vaid and Negussey [4] believed that the methods using dummy rod inclusions or assuming isotropic behavior of sand during loading were invalid, and two alternative methods were proposed for the assessment of membrane penetration. Frydman et al. carried out isotropic compression tests on full triaxial specimens and hollow cylinder specimens, and the results showed a linear relation between the measured volumetric strains [11]. The ratio of $\Delta V_m / A_m$ was used to calculate $\Delta Vm$, and a unique relation was developed between S (the slopes of lines between membrane penetration per unit area and a semilog of pressure) and the particle size for membranes with usual thickness. In addition, many investigations obtained the same conclusion, showing that the factors of membrane penetration were related to the particle size (especially the mean particle size d50), effective confining pressure, and the thickness of the rubber membrane [12–18].

The abovementioned theoretical approach was used to calculate membrane penetration. Some available methods to reduce membrane penetration are summarized below.

Method I changes the stress condition of the rubber membrane to reduce membrane penetration. For example, the surface of the penetrating membrane is coated with liquid rubber or liquid polyurethane. A thin curved brass plate or polythene strips are placed between the specimen and the membrane, and a sluice with sand is located at the periphery of the cylindrical specimen [6,12,14,19,20]. However, these methods, in essence, increase the thickness of the rubber membrane, and the membrane applies more resistance to the applied axial stress. Once lateral pressure is applied to the specimen, there is less uniformity of pressure because the stiffened membrane tends to provide strain radially during shear loading [12]. Thin, curved brass plates or polythene strips were placed between the specimen and the membrane. The overlapping portion of the brass plates was compressed by lateral pressure, and the brass plates or polythene strips offered more resistance to the penetration of the membrane into the soil voids. Sluicing the specimen with sand along the periphery of the cylindrical specimen could reduce the void space available for membrane penetration and reduce the unit membrane penetration [20,21]. However, this approach was limited to remolded specimens because sluicing of sand in undisturbed specimens changes the field conditions [22]. Method II changes the operation method of

the measurement system and improves the accuracy of the measurement system. The procedure involves the compensation of the additional volume change due to membrane penetration by injecting water into the specimen. Researchers have improved the limitations of the abovementioned systems by continuously injecting or extracting water through computer control to compensate for the measurement error caused by rubber membrane compliance under undrained conditions [7,9,16,23,24]. However, these methods were determined by the analytical relation between membrane penetration and pressure; thus, the method of injecting water into the specimen has low applicability and is difficult to apply in the laboratory. Evans [14] also proposed the double-layer membrane method to measure the penetration of rubber membranes. However, the method had many limitations in operation, and some limitations were the same as those of Method I and the method by injecting water.

Another category of membrane penetration study was based on the elastoplastic theory to deduce the analytical solution, and the relevant parameters of the semiempirical formula were calibrated by measured data. Molenkamp and Luger, Baldi and Nova, Kramer and Sivaneswaran, and JI et al. also proposed the formula of unit membrane penetration [6,24–26]. However, these formulas set the mean diameter $d_{50}$ and the confining pressure as variables, and the semiempirical parameter of formula $\varepsilon_m = d_g(Pd_g/E_m t_m)^{1/3}$ was based on different materials. The semiempirical parameter has a significant effect on the calculation results of penetration.

Based on previous studies and the existing theoretical formula, this paper presents a novel procedure to measure the membrane penetration using a newly developed multi-scale triaxial apparatus. A series of isotropic consolidation drainage (CD) tests were carried out under different confining pressures. To measure the drainage volume of each confining pressure, membrane penetration will be deduced by the proposed experimental method in this paper.

## 2. Measurement of the Membrane Penetration: A Novel Procedure

### 2.1. Basic Principle

For isotropic CD tests, the total volume change of the coarse granular specimen is composed of two parts: the volume deformation of the soil skeleton and the volume change caused by membrane penetration under the specific stress state conditions. In order to measure membrane penetration of coarse granular materials, this paper starts from the basic theory of soil deformation. Figure 1 shows the relation curves of volume change versus the confining pressure, which has been widely recognized in previous researches [6,13]. The corresponding formula [26] may be expressed as:

$$\Delta V_T(P) = \Delta V_s(P) + \Delta V_m(P), \tag{1}$$

where $P$ is the confining pressure, $\Delta V_T(P)$ [mL] is the total volume change under special consolidation confining pressures, $\Delta V_s(P)$ [mL] is the volume change of the soil skeleton which is the real volume change of the sample, and $\Delta V_m(P)$ [mL] is the volume change caused by membrane penetration.

The volume change of soil skeleton can be expressed as:

$$\Delta V_s(P) = \varepsilon_s(P)V_0 \tag{2}$$

The membrane penetration can then be written as:

$$\Delta V_m(P) = \varepsilon_m(P)A_m \tag{3}$$

where $V_0$ [cm$^3$] is the initial volume of the specimen, and $\varepsilon_s(P)$ [cm$^3$/cm$^2$] and $\varepsilon_m(P)$ [cm$^3$/cm$^2$] are the soil volumetric strain and the membrane penetration per unit membrane surface area under special confining pressures, respectively. $A_m$ [cm$^2$] is the specimen surface area covered by the membrane.

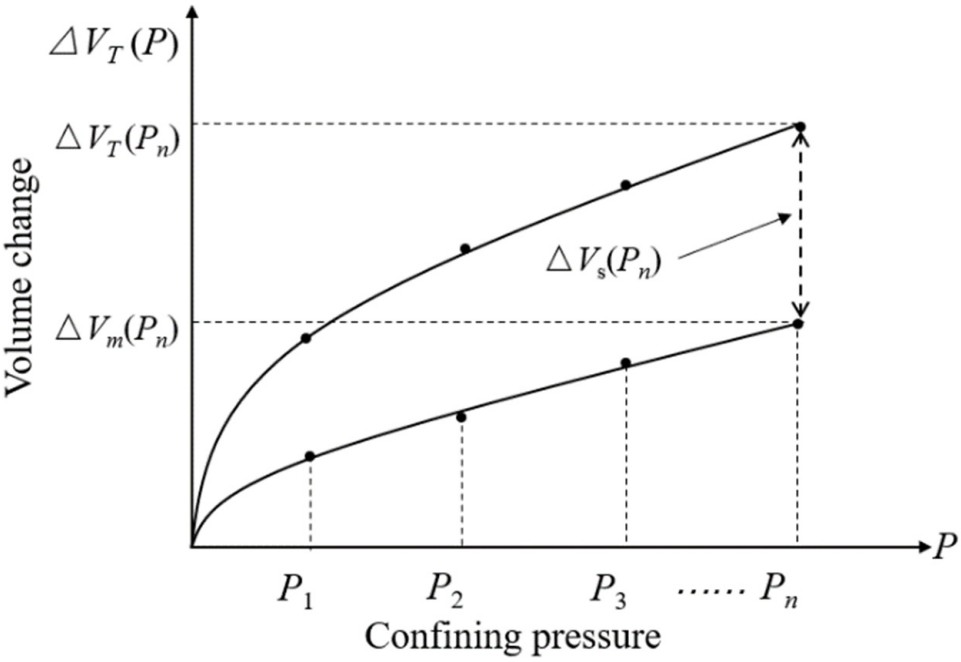

**Figure 1.** Relation curves of volume change versus the confining pressure.

Equation (1) can be rewritten as

$$\begin{aligned} \Delta V_T(P) &= \tfrac{1}{4}\varepsilon_s(P)\pi D^2 H + \varepsilon_m(P)\pi DH \\ &= \tfrac{1}{4}\varepsilon_s(P)A_m D + \varepsilon_m(P)A_m \end{aligned} \tag{4}$$

where $D$ [mm] is the diameter of the specimen and $H$ [mm] is the height of the specimen.

In this paper, the proposed method to measure membrane penetration of coarse granular materials can be expressed as

$$\frac{\Delta V_T(P)}{A_m} = \frac{1}{4}\varepsilon_s(P)D + \varepsilon_m(P) \tag{5}$$

In Equation (5), different confining pressures are applied to the peripherals of multiple specimens to measure the total volume of drainage $\Delta V_T(P)$ [mL], where $\Delta V_T(P)/A_m$ [cm$^3$/cm$^2$] is set as the vertical coordinate, and $D$ [mm] is the horizontal coordinate. The relationship between the specimen diameter and volume change can be established by extending the lines intersecting the vertical axis, and the intercept of the vertical coordinate axis is the membrane penetration per unit area, as shown in Figure 2.

Compared with previous methods requiring the application of brass rod in the central of specimen, this method can effectively transmit the hydrostatic pressure onto the soil following the vertical direction. Therefore, when the top loading platen rests on the brass rod, the undesirable non-hydrostatic compression of the soil skeleton can be averted. The rigid inclusion also restricts the axial deformation of specimen, so that the unreasonable assumption of isotropic ($\varepsilon_v = 3\varepsilon_a$) of granular materials under hydrostatic pressure which is underlain in previous methods is no longer necessary. Following this method, the membrane penetration of coarse granular materials can be derived theoretically with a series of isotropic CD tests on specimens of different diameters.

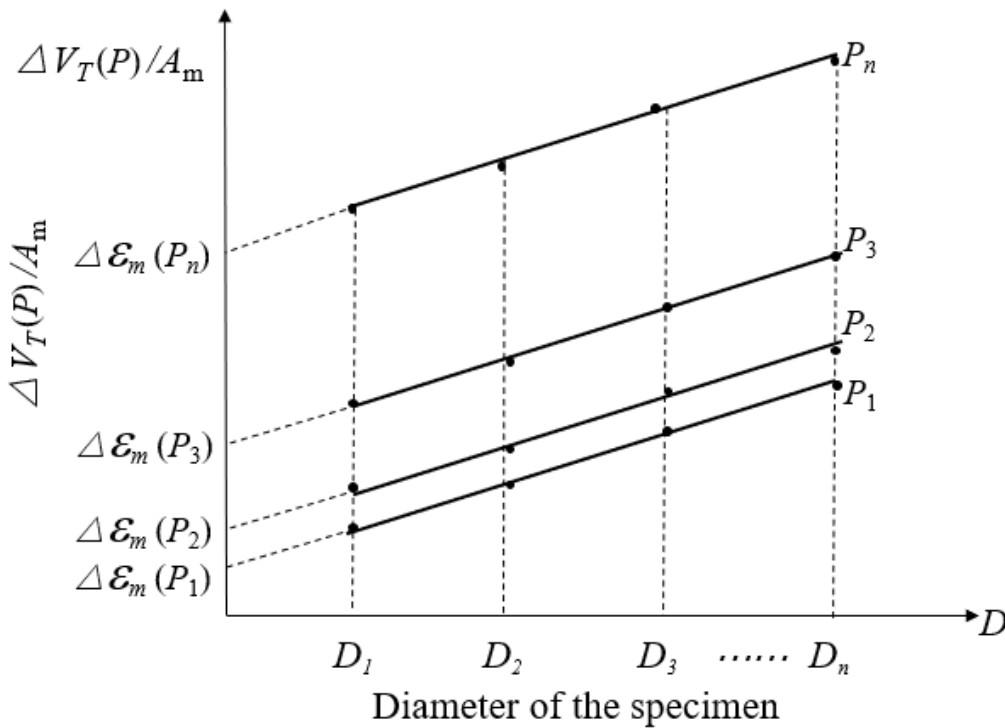

**Figure 2.** Relationship between $\Delta V_T(P)/A_m$ and the diameter of the specimen.

### 2.2. Chassis Apparatus of the Triaxial Test

Different measurement systems can significantly affect the accuracy of the triaxial tests. Due to the difference of apparatus size and loading system, the reliability of measurement results can be directly affected. For example, when the confining pressure is loaded, the specimen size is different in the increment of confining pressure loading. This will affect the drainage volume of the sample at the initial stage and have a great influence on the volume change of the sample. When the sample cannot be drained in time, it may produce pore water pressure and affect the test results. Therefore, in order to diminish the measurement error and to carry out triaxial tests of different diameters on the same triaxial apparatus, a chassis device for multiscale triaxial test apparatus, designed by Nanjing Hydraulic Research Institute, is shown in Figure 3. The chassis apparatus consists of a confining pressure import, a saturated water inlet, and a drain hole (measure the volume change of specimen). Shaped like a tower, this apparatus is divided into five layers. Each layer can be assembled into a specimen. The diameter-to-height ratio of the specimen is 0.5, and customized caps are designed to individually match the top of each specimen. A drain hole, located on the upper side of the specimen cap, is connected to the lower pipe of the drainage system. The bottom connection device of the chassis device is designed according to the size of existing pressure chamber, which can perfectly match the existing pressure chamber system. In this way, each specimen could be loaded in the same confining pressure increment. A proper confining pressure increment can ensure that each sample does not produce pore water pressure due to the excessive loading speed. The measurement error caused by different instrument size under different loading conditions is also handled in this apparatus. By installing rubber membrane on each corresponding size base and then filling the test materials, the isotropic consolidation drainage test can be carried out. When the triaxial test with a particular specimen size is finished, the base corresponding to the specimen size (for example, base 100 indicates sample diameter of 100 mm) can be removed, and the test with the following specimen size can be carried out.

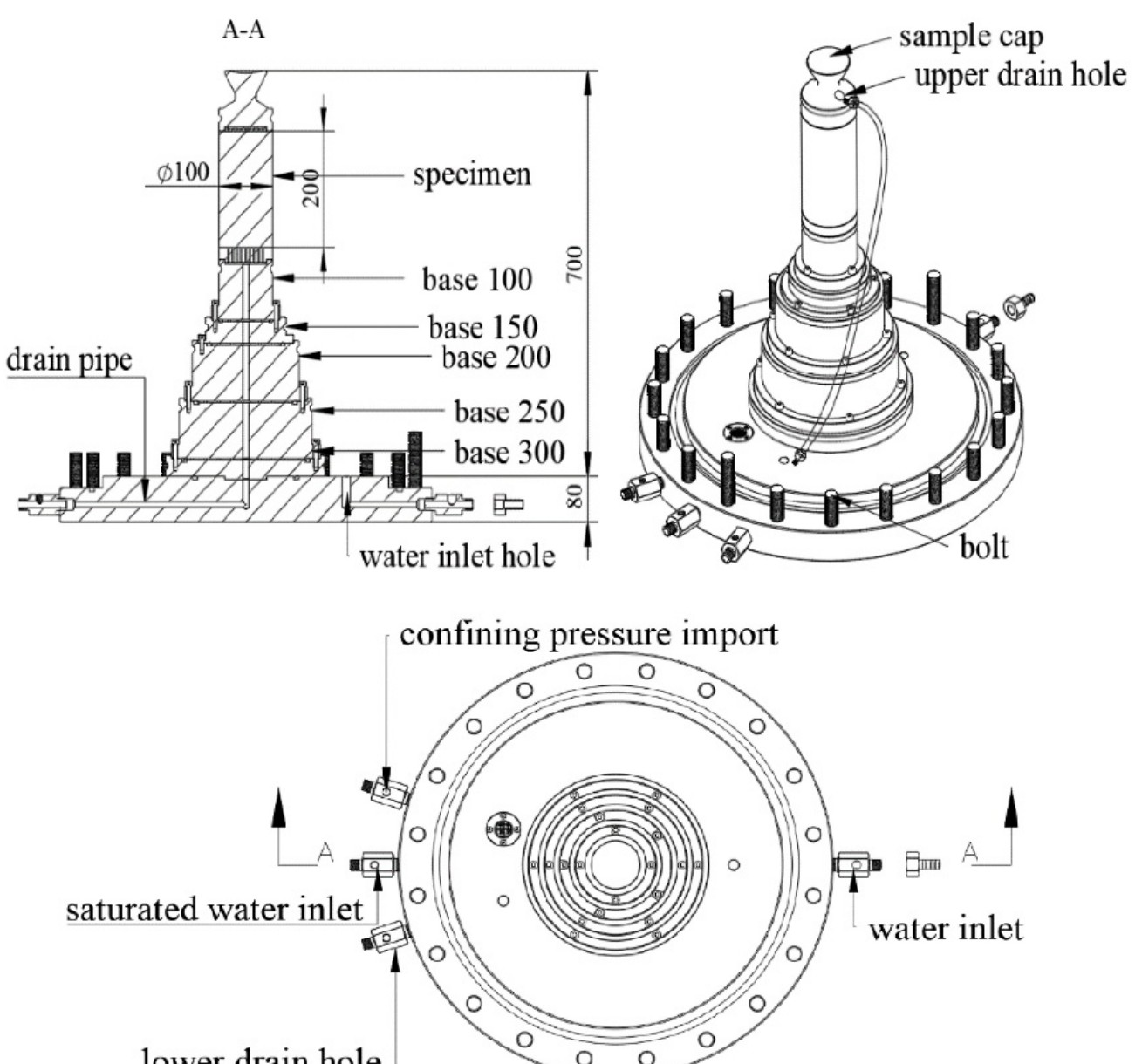

**Figure 3.** Chassis apparatus of the triaxial test.

## 3. Materials and Methods

### 3.1. Testing Material

According to ASTM D2487-17e1, Standard Practice for Classification of Soils for Engineering Purposes, the physical properties and the grain size distribution curves of the testing materials are summarized in Table 1 and Figure 4, respectively. The original coarse-grained cohesionless soil used in all proceeding tests was extracted from Sichuan and Tibet, China. It is mainly composed of gravel and sand, rockfill material, respectively. The particle size range of testing gradation curve 1 is from 0.5 mm to 10 mm, and the median particle size (d50) is 2.84 mm, the material is subject to rockfill materials (RM). The particle size range of testing gradation curve 2 is from 1 mm to 20 mm, and the average particle size (d50) is 5.98 mm. Another set of grain sizes of test soil is also in the range of testing grading curve 1 from 0.5 mm to 10 mm, and the material is subject to sandy gravel (SG1 and SG2). The thickness of the rubber membrane is $t_m$ = 2 mm, and the elastic modulus of the membrane is 1.608 MPa.

**Table 1.** Physical properties of the test materials.

| Materials | Specific Gravity $G_s$ | Initial Void Ratio e | $d_{50}$ [mm] | $\rho_d$ [g/cm$^3$] |
|---|---|---|---|---|
| RM | 2.70 | 0.59 | 2.84 | 1.70 |
| SG1 | 2.69 | 0.51 | 2.84 | 1.78 |
| SG2 | 2.69 | 0.51 | 5.98 | 1.78 |

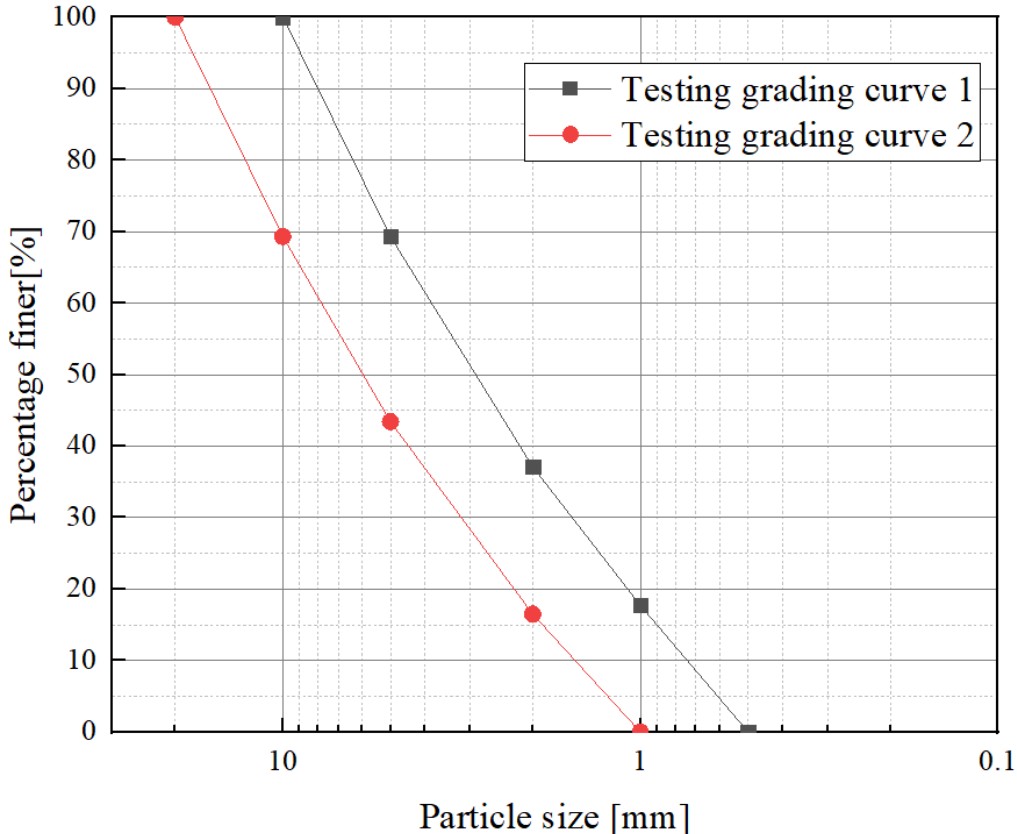

**Figure 4.** Particle size distributions of the testing materials.

*3.2. Specimen Prepration*

The volume change was measured in consolidated drained triaxial tests, and the confining pressures were loaded from 100 kPa to 600 kPa. During sample preparation, the membrane was fixed on the chassis, and the coarse granular materials were filled into the membrane in five layers. When the test with the current specimen is finished, a series of instrument operations, involving the dismantle of chassis of the current diameter and the installation of specimen with the desired dimension in the next stage, would be performed to proceed the experimental process. After the specimen preparation, a vacuum of 15 kPa was applied to the specimen, and then the specimen cap was tightly installed. The upper and lower drainage pipes were connected, the pressure chamber was covered, and then water was injected into the pressure chamber. When the water continuously and steadily flows from the upper drain outlet, the upper drain outlet valve was closed and the specimen was saturated under the water head of 2000 mm. After the completion of saturation process, the specimen was loaded with a confining pressure of 20 kPa to ensure that the rubber membrane adheres to the specimen. At this time, the value of volume changes was set back to 0. Then, the specimen was loaded from 100 kPa to 600 kPa, and the volume change of the specimen was recorded under different confining pressures. The specimens of each diameter are shown in Figure 5.

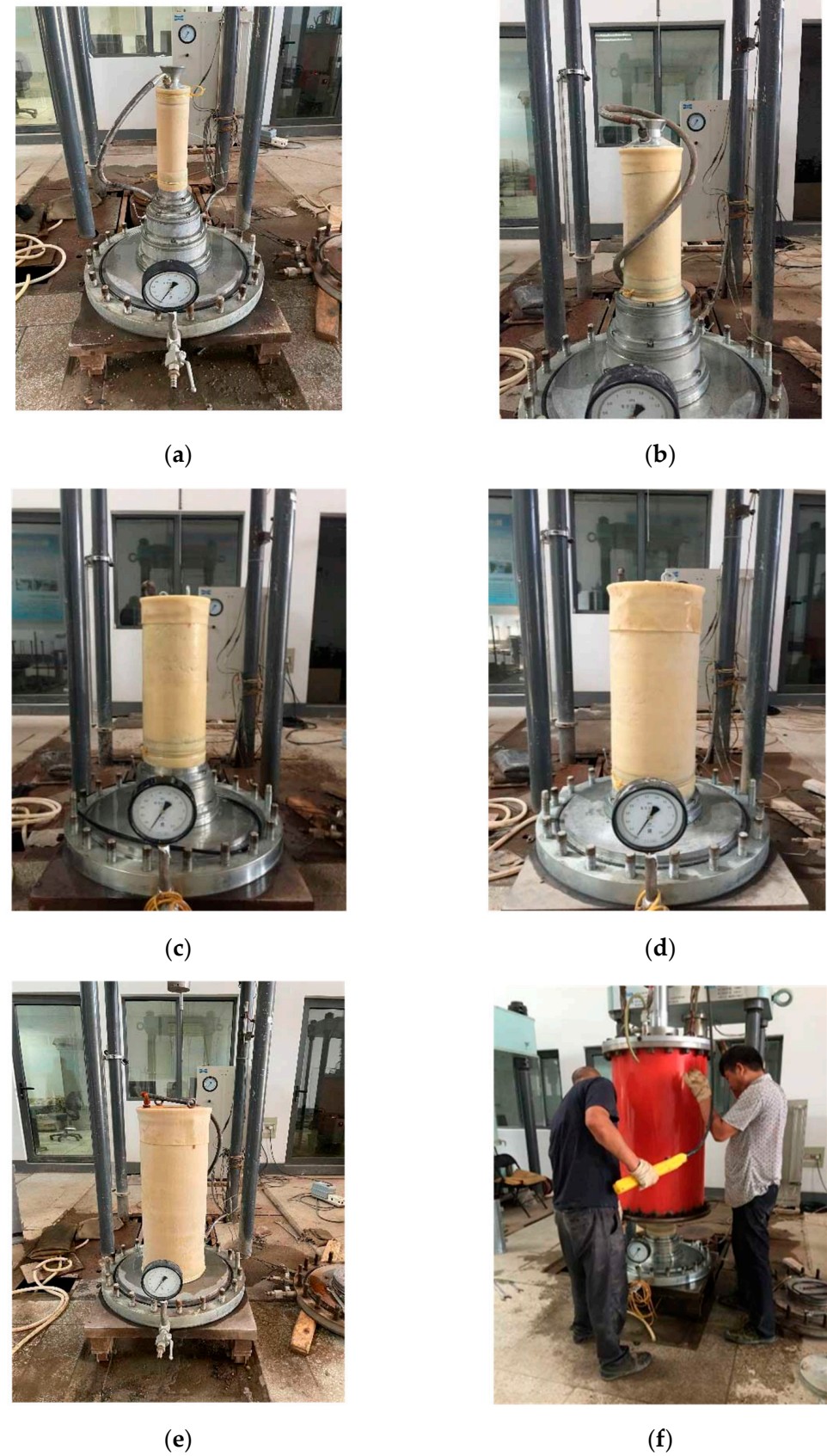

**Figure 5.** Pictures of the triaxial test: (**a**) diameter of 100 mm, (**b**) diameter of 150 mm, (**c**) diameter of 200 mm, (**d**) diameter of 250 mm, (**e**) diameter of 300 mm, (**f**) triaxial pressure chamber.

### 3.3. Test Results and Analysis

A series of isotropic consolidation drained (CD) tests of RM and SG were conducted by the aforementioned triaxial apparatus. According to the results of the isotropic CD test, the relationship between the confining pressure and the volume of drained water under different diameters is shown in Figure 6a–c. It can be seen that, with the increasing confining pressure, the measured volume change showed a hyperbolic increase. The larger the diameter, the more obvious the measured volume change increases. This is because the contact area between the specimen and rubber membrane increases significantly with the diameter of specimen increasing. With the increasing confining pressure, the penetration depth of the rubber membrane around the specimen increases, thus the measured volume change increases. Under the same gradation curve conditions, the volume change of RM is much greater compared to that obtained in test SG2. This can be attributed to the different lithology of coarse-grained soil. Meanwhile, as shown in Figure 6b,c, as the particle size increases, the measured volume change of the sandy gravel increases.

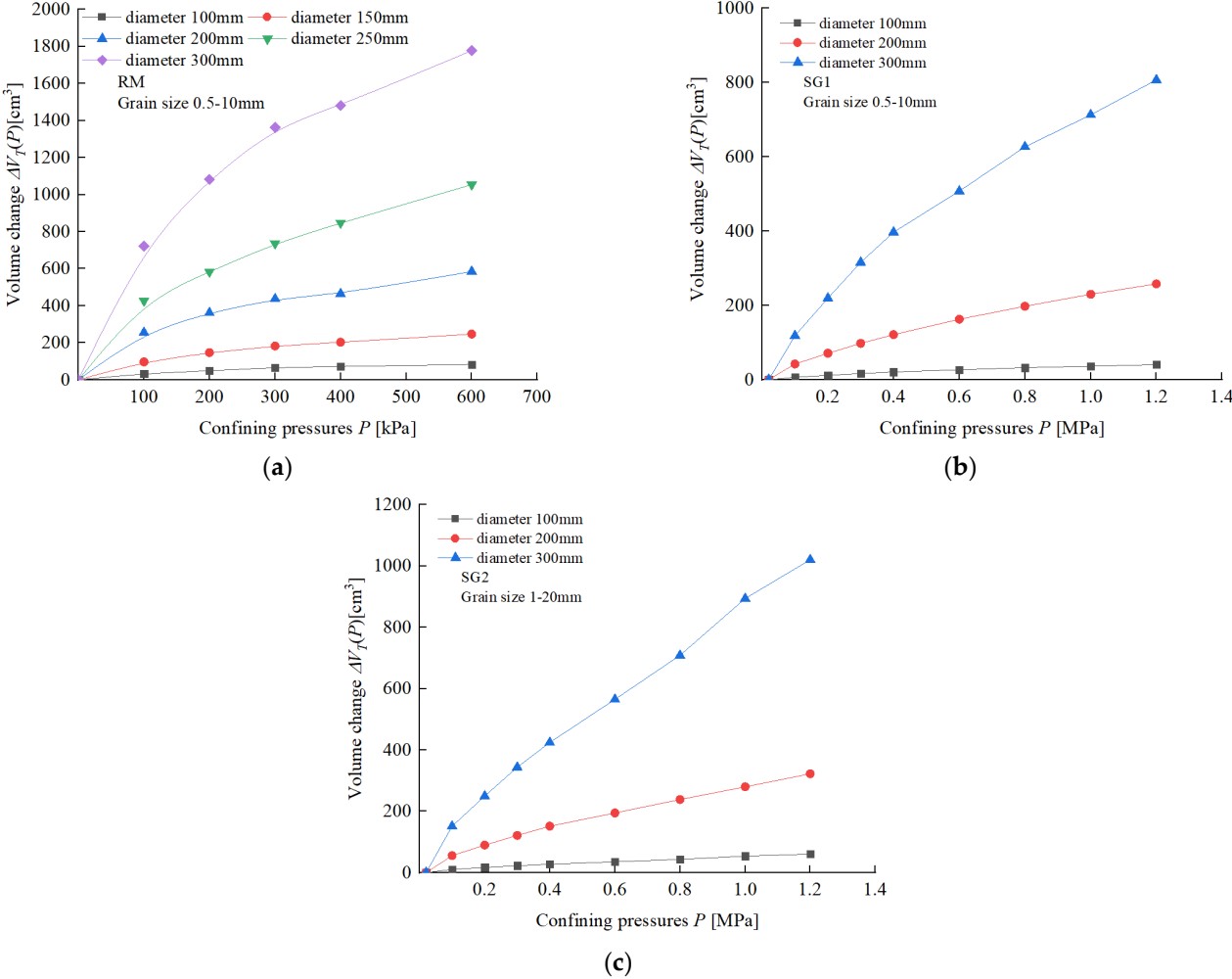

**Figure 6.** Measured volume change under different confining pressures: (**a**) RM, (**b**) SG1, (**c**) SG2.

According to Equation (1), the relationship between $\Delta V_T(P)/A_m$ and the diameter $D$ can be established (as shown in Figure 7a–d). It is found that, with the increasing confining pressure, the ratio of $\Delta V(P)/A_m$ increases, and there is a good linear relationship between $\Delta V(P)/A_m$ and $D$. Note that, in Figure 7, the fitted line is extended to intersect the vertical axis, and the intercept of the fitting straight line is the unit membrane penetration. Consequently, the unit membrane penetration under different confining pressures can be calculated.

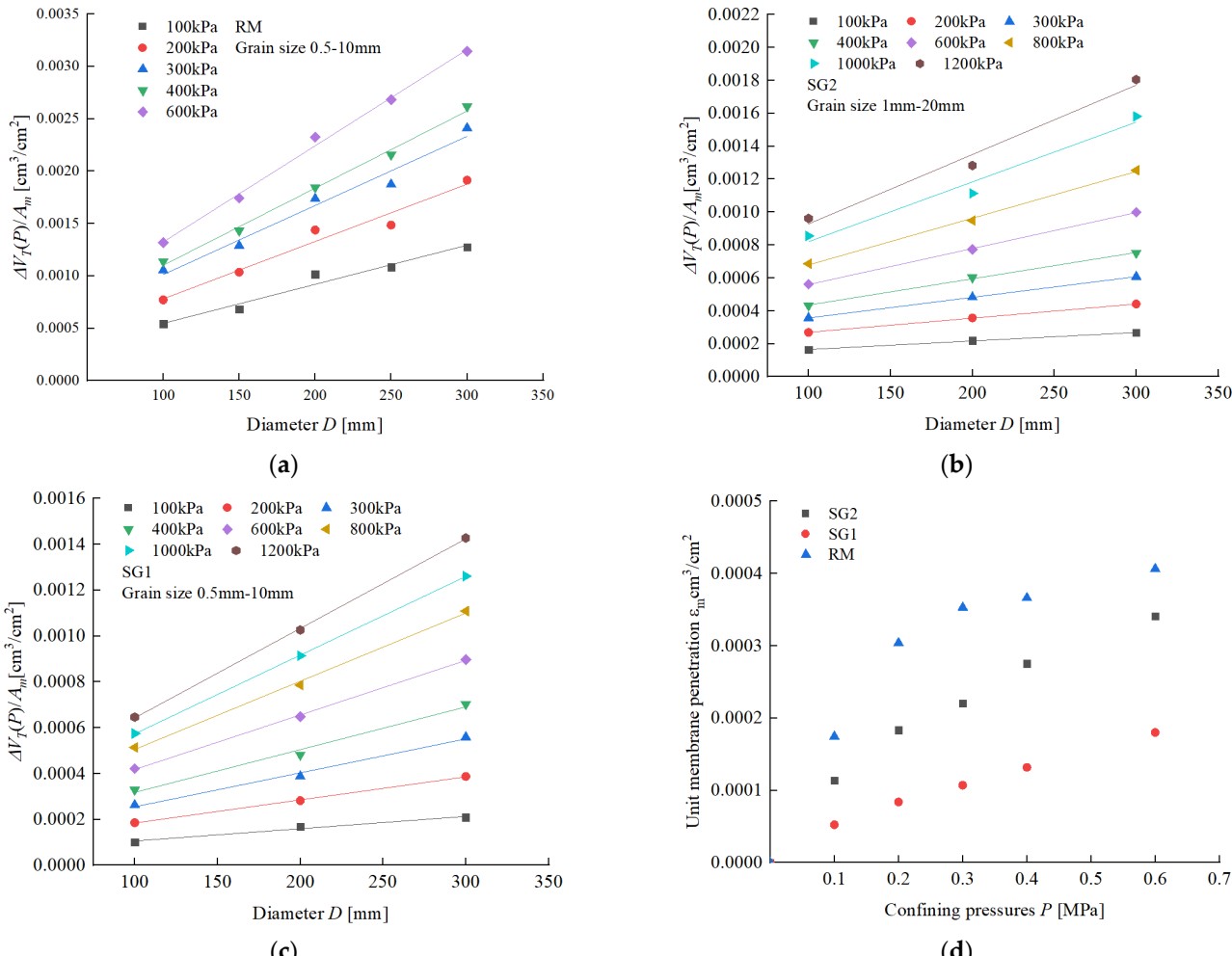

**Figure 7.** Membrane penetration from multiple triaxial specimens during hydrostatic loading: (**a**) formula fitting(RM), (**b**) formula fitting (SG2), (**c**) formula fitting(SG1), (**d**) comparison of unit membrane penetration.

According to the calculation method of membrane penetration proposed in this paper (Equation (5)), the ratio of membrane penetration to the total volume change for rockfill materials and sandy gravel under different pressures and grain sizes are shown in Tables 2 and 3.

**Table 2.** Proportion of membrane penetration for RM (%).

| RM | Diameter [mm] | | | | |
|---|---|---|---|---|---|
| **Confining Pressure (kPa)** | **100** | **150** | **200** | **250** | **300** |
| 100 | 32.1 | 25.6 | 17.2 | 16.1 | 13.7 |
| 200 | 39.4 | 29.4 | 21.1 | 20.5 | 15.9 |
| 300 | 33.5 | 27.4 | 20.3 | 18.8 | 14.6 |
| 400 | 32.1 | 25.6 | 19.8 | 17.0 | 14.0 |
| 600 | 30.8 | 23.3 | 17.5 | 15.1 | 13.0 |

Regarding the influence of membrane penetration of RM, it can be seen in Table 2 that the proportion changes slightly as the confining pressure increases, and has a tendency to decrease when the diameter is the same. However, under the same confining pressure, the membrane proportion decreases sharply. Compared with the specimen with a diameter of 100 mm, the proportion of membrane penetration with a diameter of 300 mm decreases from 32.1% to 13.7%, the proportion is reduced by 57.5% on average in the range of pressures from 100 kPa to 600 kPa. The results show that the diameter of the sample has a great

influence on the membrane penetration. It is recommended to use specimens of larger size to reduce the proportion of membrane penetration in the triaxial test.

**Table 3.** Proportion of membrane penetration for SG (%).

| SG | Diameter [mm] | | | | | |
|---|---|---|---|---|---|---|
| | 100 | | 200 | | 300 | |
| Confining Pressure [kPa] | SG1 | SG2 | SG1 | SG2 | SG1 | SG2 |
| 100 | 51.6 | 69.4 | 31.2 | 51.8 | 25.1 | 42.6 |
| 200 | 45.1 | 68.2 | 29.7 | 51.4 | 21.6 | 41.6 |
| 300 | 40.8 | 64.9 | 27.6 | 47.9 | 19.2 | 38.1 |
| 400 | 40.0 | 63.8 | 27.3 | 45.7 | 18.8 | 36.7 |
| 600 | 42.7 | 60.7 | 27.8 | 44.1 | 20.1 | 34.1 |
| 800 | 40.5 | 57.6 | 26.5 | 41.7 | 18.8 | 31.5 |
| 1000 | 40.1 | 53.5 | 25.2 | 41.2 | 18.3 | 28.9 |
| 1200 | 39.1 | 52.7 | 24.6 | 39.6 | 17.7 | 28.1 |

Meanwhile, the proportion of membrane penetration of SG is larger than that of RM, as can be seen in Table 3. The subsequent loading of confining pressure will lead to the rearrangement of internal particles, so that voids among large particles are filled with fine particles, and the samples are compacted. With the increase of confining pressure, the proportion of membrane penetration decreases gradually. By comparing test results in Table 2, it can be seen that the membrane penetration effect with RM is less affected by confining pressure than that in the case of SG. This is because the voids within RM are not easily filled due to the particle occlusion of rockfill materials and the lack of fine particles. In addition, the proportion of SG1 is smaller than that of SG2 under the same pressure conditions. For example, when the confining pressure is 100 kPa, the proportion of membrane penetration increases from 51.6% to 69.4%. The reason is that the membrane penetration effect increases with the increase of particle size under the same confining pressure. The d50 of SG2 is 4.28 mm, which is greater than that of SG1. Similar to rockfill, with the increase of sample diameter, the proportion of membrane penetration in sandy gravel decreases gradually. This phenomenon is consistent with research findings by Baldi and Nova [25].

## 4. Discussion

### 4.1. Comparison with Previous Formulas

The semiempirical formulas for calculating the unit membrane penetration proposed by Molenkamp and Luger, Baldi and Nova, Kramer and Sivaneswaran and Ji et al., are shown in Equations (6)–(9), [6,24–26].

$$\text{Molenkamp and Luger}: \; \varepsilon_m = 0.16 d_g \left( \frac{P d_g}{E_m t_m} \right)^{\frac{1}{3}} \tag{6}$$

$$\text{Baldi and Nova}: \varepsilon_m = 0.125 d_g \left( \frac{P d_g}{E_m t_m} \right)^{\frac{1}{3}} \tag{7}$$

$$\text{Kramer and Sivaneswara}: \; \varepsilon_m = 0.231 d_g \left( \frac{P d_g}{E_m t_m} \right)^{\frac{1}{3}} \tag{8}$$

$$\text{JI et al.}: \; \varepsilon_m = 0.66 d_g (1+e)^{\frac{4}{9}} \left( \frac{1-\alpha}{4M} \right)^{\frac{1}{3}} \left( \frac{P d_g}{E_m t_m} \right)^{\frac{1}{3}} \tag{9}$$

where $a = 0.732 d_g (1+e)^{\frac{1}{3}}$, $\alpha = 0.15 \left( \frac{Pa}{E_m t_m} \right)^{\frac{1}{3}}$, and $M = 324.7\alpha^4 + 237.3\alpha^2 - 3.5\alpha + 20.2$.

Based on these semiempirical formulas, the relationships between the unit membrane penetration and confining pressure are shown in Figure 8a–c.

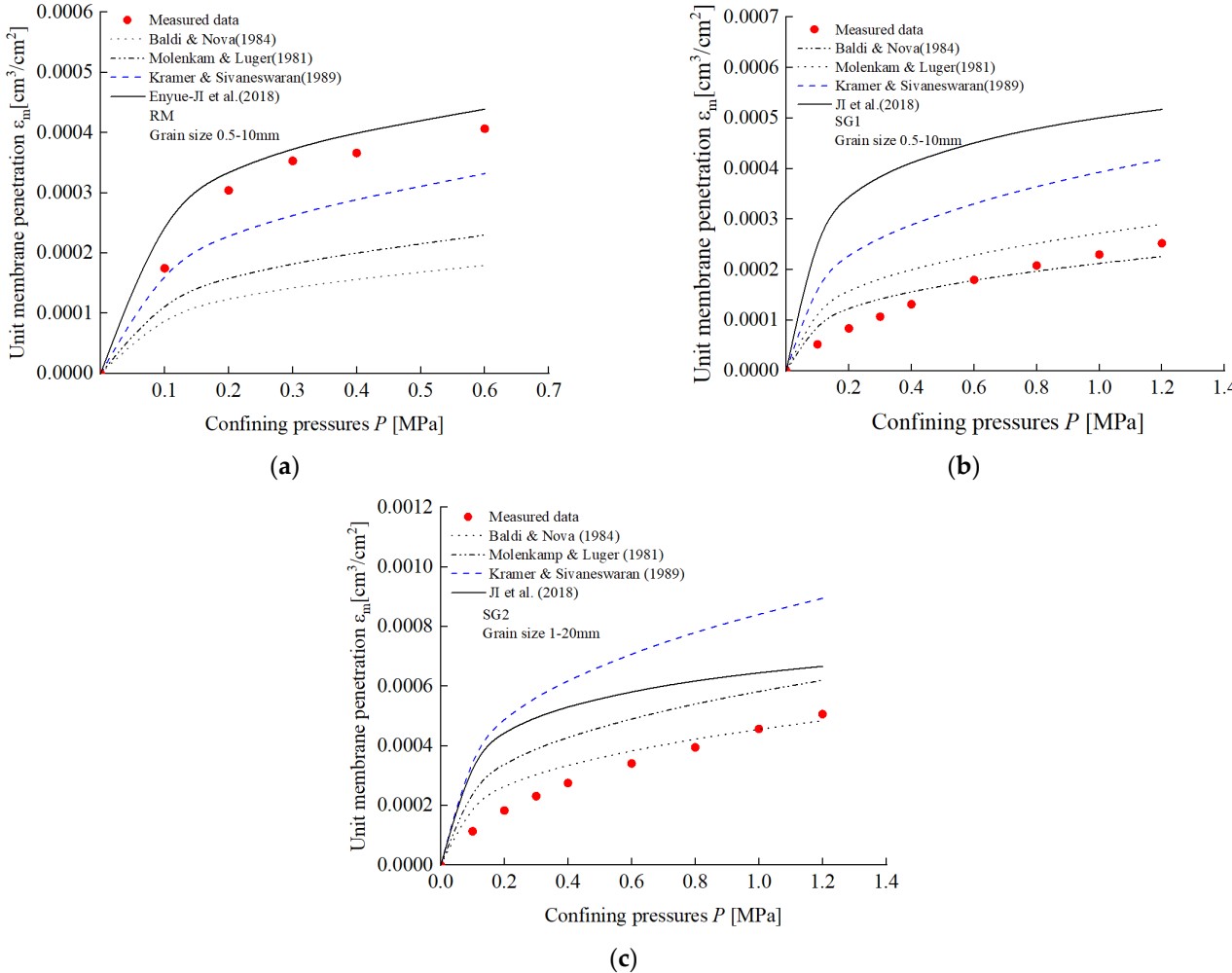

**Figure 8.** Relation curves of unit membrane penetration versus confining pressures: (**a**) RM, (**b**) SG1, (**c**) SG2.

For the rockfill materials, the overall trends of membrane penetration calculated by the semiempirical formulas are the same (see Figure 8a). In other words, the unit membrane penetration increases with the increasing confining pressure. Meanwhile, the calculation results of Equations (6)–(8) are smaller than the test results, and the calculation result of Equation (8) is close to that of the measured data. Figure 8b,c shows relationships between the experimental unit membrane penetration and the calculated value of SG considering the changes of confining pressure. In contrast to other methods, the results of the semiempirical formula proposed by Baldi and Nova are close to the experimental data. In general, the prediction results of the previous semiempirical formulas of the two materials are quite different, and this may be related to different parameters used in these formulas. The theoretical formula proposed by JI et al. is related to the median particle size $d_{50}$, pressure, and void ratio e [26]. Equations (6)–(8) are based on the median particle size $d_{50}$ and pressure. For rockfill materials, the influence of sample void ratio e on membrane penetration is much greater than that of gravel materials. Therefore, the penetration calculated by the formula proposed by JI et al. is close to the experimental value. The reason is that the edge angle of rockfill particles is clear, and the bite force and friction force between particles are large based on macroscale granulometric researches [4–6]. The particles are not easy to slide and roll under certain confining pressure, so that the fine particles in the sample cannot be filled into the voids of coarse particles in time. However, the particle shape of the sandy gravel is oval or round, so that the fine particles within sandy gravel are prone to sliding and

rolling. As a result, the cohesiveness and friction angle of sandy gravel are relatively small, and the fine particles in the sample are easy to adjust and move to the voids of coarse particles. Therefore, the void ratio of rockfill materials has a great influence on membrane penetration, and the void ratio e should be considered in the theoretical formula. On the other hand, soil particles within SG have a strong ability to self-adjust, so that void ratio has little influence on changes of membrane penetration.

*4.2. Comparison with Existing Experimental Procedures Addressing Rubber Membrane Penetration*

Existing methods to reduce the membrane penetration include the coating of polyurethane at the inner surface of membrane, the placement of thin brass plates between the specimen and the membrane, and the sand sluicing along the periphery of the cylindrical gravel specimen. However, these methods will change the stress uniformity of the specimen and the field condition. Besides, in the undrained test, it is not practical to inject water into the specimen so as to compensate for the additional volume change caused by membrane penetration, mainly because of the difficulty in experimental operation. Based on the membrane penetration tests mentioned above, the new measurement procedure proposed in this paper can be used to correct the error of volumetric deformation measurement results caused by membrane penetration in triaxial test of coarse granular materials. The chassis apparatus designed in this paper can be used to carry out the triaxial tests of specimens with different diameters in the same triaxial apparatus. Being regarded as a new experimental procedure, the integration of apparatus and methodology shows its advantage in stress application and strain measurement. This new method for measuring the membrane penetration is also verified to be of good operability and practicability in the tests of sandy gravel and rockfill materials.

**5. Conclusions**

Previous methods to evaluate the rubber membrane penetration effect involve the use of brass rods or the assumption of isotropic specimen, which do not conform to real experimental practices. To this end, a simple experimental procedure is developed for measuring the membrane penetration in this paper. A multiscale triaxial apparatus was designed, and a series of isotropic consolidation drainage triaxial tests were carried out on specimens with different diameters. The following preliminary conclusions are obtained:

(1) The penetration effect of the rubber membrane should not be ignored. The ratio of the membrane penetration to the total volume change for rockfill materials ranged from 13.0% to 39.4%, while the penetration proportion of sandy gravel reached 69.4%. Compared with the small-sized triaxial test, the large-scale triaxial consolidation drainage test can reduce the penetration proportion of the membrane and should be recommended and promoted.

(2) There is a great difference in the experimental results of the membrane penetration of rockfill materials and sandy gravel calculated by the former formula, and the former formula is not applicable to all materials. Therefore, the membrane penetration should be corrected by comprehensively considering the particle size and lithology of materials.

The method proposed in this paper and the developed apparatus has obvious advantages over previous methods in operability and practicability. The unit membrane penetration is deduced from theoretical analysis, and the method is applied to the triaxial tests of rockfill materials and sandy gravel. The equation verified in this paper can be used to calculate the membrane penetration and to further investigate the law of change under different confining pressures.

**Author Contributions:** W.X. wrote the manuscript; E.J. edited and revised the manuscript; B.S. and C.C. collected the data, drew the Figures and analyzed the data. All authors have read and agreed to the published version of the manuscript.

**Funding:** This research was supported by NHRI (Grant Number Y320008 and Y322001).

**Institutional Review Board Statement:** Not applicable.

**Informed Consent Statement:** Not applicable.

**Data Availability Statement:** Not applicable.

**Conflicts of Interest:** The authors declare no conflict of interest.

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
