# Peer review of "A Novel Procedure to Measure Membrane Penetration of Coarse Granular Materials"

_applsci, doi:10.3390/app12136381_

Round 1
Reviewer 1 Report
The paper deals with a novel procedure to measure membrane penetration of coarse granular materials. The paper corresponds to the scope of the journal. The work is original. The title is clear, concise and relevant. The abstract is representative of the work presented. Objectives are good enough stated and achieved. Graphics are clearly depicted.
The reviewer checked the plagiarism rate of the paper with Turnitin sophisticated software; with general setup the result was lower than 20%. It means that this rate can be accepted.
The reviewer recommends the following modifications:
• please give the category of the paper: “Type of the Paper (Article, Review, Communication, etc.)”; the reviewer hesitated at the first moment on it whether to which category the paper related. Please supplement this information!
• please supplement all of your figures with the title of the axes (i.e., axes of the graphs, e.g., Figs. 1 and 2),
• please make it clear which part of the title of the axes of graphs is the title and which is the unit (e.g., Figs. 6-8), the “/” sign can be considered as dividing; please put the units in brackets, e.g., “P [MPa]”, or [cm3/cm2], etc.,
• please give all the units for all the equations, because they are one of the most important part of the equations in the explanation after the equations,
• in tables, the units are missing, or they are in lower cases, please correct them, e.g., Table 1; the units are recommended to put in rectangular brackets, i.e., e.g., [g/cm3], etc.,
• please modify the units in graphs, i.e., there are missing spaces between the numbers and the units, e.g., “diameter 100 mm”, instead of “diameter 100mm”, or “grain size 0.5-10 mm” instead of “grain size 0.5-10mm”,
• in Fig. 4, the authors show the PSD curves of the granular materials; what is about the percentages between 0 and 20%? There is a missing part in the PSD curves,
• in Figs. 6 and 7, the title of vertical axis is inadequate, the “delta” sign is missing, because they are volume changes not volumes,
• the introduction part needs to be extended by discussing more relevant and up-to-date papers. The authors should appropriately extend this section by discussing more relevant works focusing on different methods and models in the literature. For example, it is suggested to read and discuss about the following relevant works:
1. Amare, M. D., & Tompai, Z. (2022). A Review on Factors Affecting the Resilient Modulus of Subgrade Soils. Acta Technica Jaurinensis, 15(2), 99–109. https://doi.org/10.14513/actatechjaur.00636
2. Alsirawan, R. (2021). Analysis of Embankment Supported by Rigid Inclusions Using Plaxis 3D. Acta Technica Jaurinensis, 14(4), 455–476. https://doi.org/10.14513/actatechjaur.00615
3. Liu, J., Sysyn, M., Liu, Z., Kou, L., Wang, P. (2022). Studying the Strengthening Effect of Railway Ballast in the Direct Shear Test due to Insertion of Middle-size Ballast Particles. Journal of Applied and Computational Mechanics, (), -. doi: 10.22055/jacm.2022.40206.3537
Questions:
• in Fig. 4, the authors show the PSD curves of the granular materials; what is about the percentages between 0 and 20%? There is a missing part in the PSD curves,
Author Response
We appreciate all constructive comments provided by the reviewer.

Reviewer 2 Report
The paper presents designed a multi-scale triaxial apparatus and a novel experimental procedure ior the measurement of unit membrane. The topic and scope correspond with journal Applied Sciences. Authors present result of a series of tri-axial tests on specimens with different diameters to study the relationship between the unit membrane penetration and total volume change.
Please consider just minor revision which can improve the paper, as follow:
- all combined citation as for example [11–17] (line 84), [5,11,13,18–19] (line 91), and [6,8,15,22–23] (line 108) should be consider more individally.
Author Response

(The authors gave the same response as above.)

Reviewer 3 Report
The manuscript discusses a newly developed procedure for measuring membrane penetration of coarse granular materials. The submitted material ideally fits 'Applied Sciences'. It presents the developed device prototype (chassis apparatus of the triaxial test - Fig. 3) and the relevant validation tests. The overall paper's form is clear, and the entire project is pertinent. Hence, I'm convinced that the material should be published. Nevertheless, while reading the text, I spotted some shortcomings needing improvement.
- Please check all cited references - above all, in the form of providing citations! The text misses spaces - e.g. lines 41,46,53,59,63,70,73,78 and so on. The remark refers to the entire paper!
- Line 70 - the word 'adaptation' means something else than 'adoption'. Please correct the sentence respectively.
- Line 130 and the following ('Basic principle'). Please cite relevant literature and express that it's fundamental knowledge if it's not your original finding.
- Figures 1 & 2 - do the figures present the authors' findings or just a common state-of-the-art? In such a case, please provide readers with the relevant explanation.
- Lines 177-180 - the fragment informs that the Nanjing Hydraulic Research Institute designed the device. Please complete the fragment with information about the premisses for elaborating on such an instrument. Why such a particular design? Are there any preceding works? Were the assumptions already published, or is it just a modification? Maybe is it a brand new solution? Please add some explaining sentences if possible.
- How did you assess the possible performance of the device? Did you execute any modelling? On which basis did you estimate the instrument's capacity?
Should you provide all necessary improvements, please re-submit the text for a final reviewing check. Good luck!
Author Response

(The authors gave the same response as above.)

Reviewer 4 Report
​General remarks
1. Article is clearly written and easy to follow but still some editing and changes especially to the abstract and introduction are required.
2. The methods, experimental and discussion parts of the paper are well written and no specific mistakes were detected.
2. The abstract requires rewriting. It is not introducing the basic overview of the paper. It is also written in a way that a person not familiar with the topic can not understand what the authors are proposing in their research. In the present form, it looks like cut out fragments of the article with some conclusions mixed in a short text.
3. Line 26-27 – according to who? No reference included.
4. Next sentence ending in line 30 also no reference to this method used by other researchers. Similarly next sentences. Please keep in mind this is an “introduction” chapter. All related statements must be connected to exemplary research. It is crucial e.g for line 45 where the authors use the strong statement “must” but do not give the reference.
5. The authors focus on a few specific technologies for membrane penetration measurements. There are of course others which do not have to be evaluated in detail but it would be a good idea to give an overall sense of what is also technically possible. The reviewer would suggest at least mentioning in the introduction some optical methods like 3D Laser Vibrometry (3D LDV) and Digital Image Correlation (DIC). In the case of 3D LDV, it can be used for non-contact tests of high accuracy during laboratory tests as well during the object operation. Here is an example of the use for light structures testing “Non-invasive measurements of ultra-lightweight composite materials using laser Doppler Vibrometry system, Guinchard, M., Proceedings of the 26th International Congress on Sound and Vibration, ICSV 2019,”. This is not specifically for example for membrane testing but the method can be also applied for such tests. DIC is also an interesting choice for testing and structure evaluation. According to my knowledge, this method was used for membrane testing at CERN.
6. Fig. 1 is it made by authors or taken from other sources. No reference is present. Fig.2, 4 similar problems.
7. Some editorial errors were detected like e.g double dots at the end of some sentences, space missing when starting new sentences etc. Additionally, it is customary to use a space between the value and the unit (except degC and %). Please make sure you change this (e.g lines 196, 212, 228, 290 check the whole article). In case of % delete the space (e.g line 291)
4. The conclusion part is well written. The authors give also an indication of the next steps and possibilities
Conclusions:
The main problem is the quality of the abstract which is problematic for someone not familiar with the field and it does not really give a clear presentation of this research. There is also a small problem at the beginning of the introduction with no reference pointing out to other research proving the statements of the authors. Thus the reviewer is asking for major reviews and will be glad to accept the paper after this and other small changes to the abstract and the introduction are made.
Author Response

(The authors gave the same response as above.)

Round 2
Reviewer 3 Report
Dear Authors,
First of all, I wish to thank you for your answers to my concerns. I appreciate your improved version; however, I still found some minor shortcomings while reading it. Please correct them before publishing:
- no matter phraseological considerations, I still believe that 'adoption' means something else than 'adaptation'. I'm aware that many authors mix it up, taking no care of the appropriate language but - at least - we should follow good practices. It's your article, so please decide whether you want it to be written correctly or commonly;
- please watch out for figures 1 and 4; there is something whimsical about their formatting;
The rest of your answers: I accept your point of view.
Wishing you good luck,
Author Response
Thank you for your kind suggestion.We have revised it according to the comments of reviewer.

Reviewer 4 Report
Dear Authors,
The manuscript was improved significantly and now it is sufficient for publication. Although it still can be improved in some aspects which are not crucial for understanding or following the research presented by the authors, the reviewer accepts the publication in the present form.
Best regards,
The reviewer
Author Response
Thank you!